# Intervention development and treatment success in UK health technology assessment funded trials of physical rehabilitation: a mixed methods analysis

Victoria A Goodwin, Jacqueline J Hill, James A Fullam, Katie Finning, Claire Pentecost, David A Richards

Institute of Health Research, University of Exeter Medical School, Exeter, UK

**Correspondence to**
Professor Victoria A Goodwin;
v.goodwin@exeter.ac.uk

## ABSTRACT

**Objectives** Physical rehabilitation is a complex process, and trials of rehabilitation interventions are increasing in number but often report null results. This study aimed to establish treatment success rates in physical rehabilitation trials funded by the National Institute of Health Research Health Technology Assessment (NIHR HTA) programme and examine any relationship between treatment success and the quality of intervention development work undertaken.

**Design** This is a mixed methods study.

**Setting** This study was conducted in the UK.

**Methods** The NIHR HTA portfolio was searched for all completed definitive randomised controlled trials of physical rehabilitation interventions from inception to July 2016. Treatment success was categorised according to criteria developed by Djulbegovic and colleagues. Detailed textual data regarding any intervention development work were extracted from trial reports and supporting publications and informed the development of quality ratings. Mixed methods integrative analysis was undertaken to explore the relationship between quantitative and qualitative data using joint displays.

**Results** Fifteen trials were included in the review. Five reported a definitive finding, four of which were in favour of the 'new' intervention. Eight trials reported a true negative (no difference) outcome. Integrative analysis indicated those with lower quality intervention development work were less likely to report treatment success.

**Conclusions** Despite much effort and funding, most physical rehabilitation trials report equivocal findings. Greater focus on high quality intervention development may reduce the likelihood of a null result in the definitive trial, alongside high quality trial methods and conduct.

## BACKGROUND

Rehabilitation is '*a set of interventions designed to optimise function and reduce disability in individuals with health conditions in interaction with their environment*'[1] and is an essential aspect of healthcare provision. By its very nature, rehabilitation in clinical practice is an individually focused, complex activity, involving interventions that are multifaceted and often

### Strengths and limitations of this study

► To our knowledge, this study is the first to use mixed methods integrative analyses to explore the relationship between quality of intervention development work and treatment success.
► Using the National Institute of Health Research Health Technology Assessment Journal monographs, published protocols and other supporting publications for each study together provided a detailed and rich source of data beyond what would be found in a single traditional journal publication.
► The study reviewed randomised controlled trials of physical rehabilitation from a single UK funder as an exemplar and therefore findings may not be representative of other complex interventions or other funding bodies.

implicit in nature,[2] and as such, historically, this has been viewed as a barrier to undertaking research.[3] This said, there is a growing body of randomised controlled trials (RCTs) of rehabilitation, suggesting that these challenges can be overcome.[4] This may, in part, be supported by the publication of the Medical Research Council (MRC) Framework for developing and evaluating complex interventions.[5 6]

The MRC framework was developed to optimise the likelihood that new interventions are not rejected as being ineffective when inadequate effort has been made in the development of the intervention.[7] Likewise, Chalmers and Glasziou[8] highlighted the importance of avoiding research waste and recommended that sufficient effort is made to ensure the relevant research questions are identified and addressed using high quality research methods. However, there appears to have been no formal evaluation of the impact of using the development component of the framework on trial outcomes and whether we

are observing evidence of effective interventions being developed.

Previous UK[9] and US[10] reviews synthesised successful and non-successful treatment outcomes from trials of new interventions in order to assess the equipoise principle and to understand what return has been achieved on the investment made by those taking part in the trials, researchers and funders. Dent and Raftery[9] reported 24% (20/85) primary outcome comparisons as having a positive result, of which 19% (16/85) were in favour of the new intervention, with 22% (19/85) comparisons reporting a true negative outcome. However, these authors did not focus on rehabilitation interventions, nor did they seek to understand factors that may impact on treatment success, such as the quality or intensity of intervention development pretrial procedures. Informal discussions with colleagues in the UK and internationally noted that an increasing number of publicly funded, large RCTs evaluating physical rehabilitation interventions had reported null findings. Similar concerns have been reported in studies of public health interventions.[11][12] Our study, therefore, sought to assess this observation and also explore whether intervention development activities contributed to treatment success using the National Institute of Health Research Health Technology Assessment programme (NIHR HTA) as an exemplar.

We aimed to use data from the NIHR HTA to

a. Establish the treatment outcomes of funded RCTs of physical rehabilitation.
b. Establish how many new interventions were found to be effective.
c. Examine what work had been done in terms of developing the new intervention.
d. Examine the relationship between (a) and (c).

We adopted a mixed methods approach to address the study aims. Although evidence of using integrative mixed methods approaches in synthesising evidence on complex interventions is limited, mixing together qualitative and quantitative data can generate understanding that has the potential to be greater than the sum of the individual parts.[13]

## METHODS
### Design
We undertook a review of NIHR HTA funded RCTs of physical rehabilitation interventions using narrative synthesis of outcomes and mixed methods analysis of the relationship between intervention development and categorical treatment outcomes using joint displays.

### Patient and public involvement
Patients and the public were not involved in this study.

### Data sources and inclusion criteria
We included superiority RCTs of physical rehabilitation funded by the NIHR HTA programme. The interventions could be delivered by a single profession or be multiprofessional. The NIHR HTA programme is the leading public funding source for RCTs in the UK and trials of rehabilitation are increasingly part of the portfolio. We only included completed RCTs whose main trial findings were reported in an HTA monograph or peer-reviewed publication in order to establish treatment success. We excluded pilot and feasibility RCTs as they do not aim to assess the efficacy or effectiveness of interventions[14]; studies where the interventions were primarily psychological or cognitive as the focus of the study was physical rehabilitation; where the primary outcome findings were not reported with a 95% CI as these data were required to assess treatment success.

### Search and screening
We searched the HTA Project Portfolio (since superseded by the NIHR Journals Library https://www.journalslibrary.nihr.ac.uk/#/) from inception to July 2016 using the following keywords: physiotherap*OR occupational therap* OR speech and language therap* OR rehabilitation. We removed duplicate, and then titles and scientific abstracts were reviewed for potential inclusion by one person and checked by a second. Subsequently full text reports were screened for inclusion by one person and checked by a second. Any disagreements were discussed and agreed with a third person.

### Data extraction
All data were extracted by one person and checked by a second. Discrepancies were discussed and resolved with a third person.

#### Quantitative trial data
Data extracted from each trial publication included trial design, target population, health categories (using the Health Research Classification System), primary outcome(s) and time point, minimal important clinical difference or percentage change that the trial aimed to detect, planned and achieved sample size, and primary outcome results with 95% CI. We also recorded the professional background of the chief investigator and amount of funding awarded.

#### Qualitative intervention development data
Using the revised version of Criteria for Reporting the Development and Evaluation of Complex Interventions (CReDECI 2)[15] and the Template for Intervention Description and Replication checklist (TIDieR)[16] as frameworks, we extracted all available documentary (qualitative) data from the body of the text regarding intervention development, including descriptions of underlying theory, intervention components and reasons for selection, intended interactions between components, contextual considerations, piloting of intervention and impact of definitive intervention to be evaluated, control components, planned intervention delivery and materials. Where additional supporting publications were cited, such as a protocol or intervention development studies, we used these as additional sources of documentary data.

## Data analysis

We used summary statistics to describe the characteristics of the included studies. We categorised primary outcome findings into one of six treatment outcome categories as described by Djulbegovic and colleagues,[10] these being (1) statistically significant in favour of the new treatment, (2) statistically significant in favour of the control treatment, (3) true negative, (4) truly inconclusive, (5) inconclusive in favour of new treatment or (6) inconclusive in favour of the control treatment. This was achieved by comparing the 95% CI for the difference in primary outcome to the difference specified in the sample size calculation.[9] If the 95% CI excluded a meaningful difference in either direction, implying the treatments have similar effects, the results were categorised as true negative. If the 95% CI included a meaningful difference in either direction (ie, trial failed to answer the primary question), the results were categorised as being truly inconclusive.

Where a single primary outcome and primary time point were not explicitly identified, we used the following hierarchy to determine which primary outcome would be used in the analysis:

▶ Explicitly defined primary outcome.
▶ Outcome used in power calculation.
▶ Main outcome stated in trial objectives.
▶ First outcome reported in sample size calculation.

If a primary time point was not reported, we used the first follow-up time point as this is when we would expect the intervention to have had the greatest effect.

Our preliminary analysis of the qualitative documentary data involved the reading and re-reading of source documents and the extracted descriptions to consolidate our understanding of the development work undertaken in each study. Using a reflective and iterative process, we undertook thematic analysis to distill, structure and make sense of intervention development activity by coding and organising data into themes and subthemes. Each theme and subtheme provided a coherent description of the development work undertaken for each study, which were then synthesised into short descriptors to allow us to produce summary tables. The summary tables comprised a row for each study with columns for each theme and, where relevant, each subtheme. A second researcher checked, discussed and refined descriptors to ensure accuracy. From these descriptions, we then developed descriptive ratings on the quality of the intervention development. Depending on the nature of the data, ratings were categorised and the iterative process involved two researchers refining and checking ratings to ensure they reflected the summary data from each study. In order to provide a visual representation of the quality of intervention development work, these ratings were then converted to a quality coding to indicate high quality, some or unclear quality or limited quality. For example, under co-design the highest quality rating was given when the intervention was co-designed with *both* clinical and service user input, a middle rating when *either* clinicians or service users were involved, and the lowest

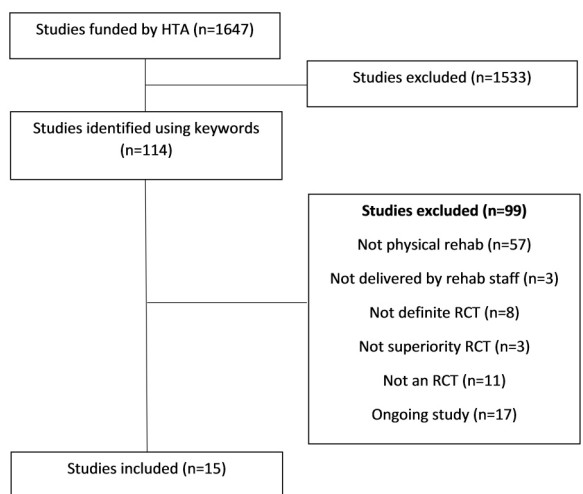

**Figure 1** Study selection. HTA, Health Technology Assessment; RCT, randomised controlled trial.

quality rating when *neither* clinicians nor service users were involved.

To examine the relationship between intervention development and treatment success, we applied mixed methods analytical techniques in novel ways. For each study, we combined ratings derived from the qualitative data on intervention development with the quantitative data on treatment outcomes in a joint display.

## RESULTS

We included 15 RCTs (figure 1),[17–31] of which 13 used a two-arm, parallel RCT design, one was a two-arm cluster RCT and one was a four-arm factorial design (of which only two arms related to physical rehabilitation). Table 1 provides a summary of the population, intervention, control and outcomes for each study. The combined sample size required to demonstrate a true difference in primary outcomes (excluding any inflation to account for loss to follow-up) was 7548 participants. The total number of participants who provided primary outcome data was higher than this (n=7834), likely due to lower loss to follow-up that estimated, although three studies[19 29 30] were considerably below their target sample size at the primary time point. Five primary outcomes were symptom-based or clinical outcomes, seven were functional measures, two were combined measures and one assessed quality of life. Primary time points varied from immediately postintervention to 1 year (median 6 months). The health categories were stroke (n=4), neurological conditions (n=2), inflammatory/immune system disorders (n=2), respiratory (n=1), musculoskeletal (n=1), cardiovascular (n=1), mental health (n=1), accident/injuries (n=1), renal/urogenital (n=1) and other (n=1). Seven interventions were delivered by physiotherapists, one by occupational therapists, one by speech and language therapists, one by nurses, two could be delivered by either a physiotherapist or a nurse, two could be delivered by a physiotherapist or an occupational therapist and one was

**Table 1** Summary of included studies

| Author (year published) | Funding awarded (£) | Population (target sample size/number of participants with primary outcome data) | Intervention | Control | Primary outcome (MCID or % change study aimed to detect) |
|---|---|---|---|---|---|
| McCarthy et al (2004)[25] | 218 517 | People with knee osteoarthritis (n=152/200) | Twice weekly exercise group for 8 weeks plus home exercises | Home exercises | Aggregate Locomotor Function score (4 s) |
| Vickers et al (2004)[28] | 161 532 | People with chronic headache (n=288/301) | Up to 12 acupuncture treatments plus usual care | Usual care from general practitioner | Weekly headache score (35% reduction) |
| Epps et al (2005)[19] | 152 011 | Children with juvenile arthritis (n=200/74) | 8 hydrotherapy and 8 land-based sessions over 2 weeks followed by weekly/fortnightly hydrotherapy for 2 months | 16 land-based exercise sessions over 2 weeks followed by weekly or fortnightly land-based exercise sessions | Disease status calculated from Childhood Health Assessment Questionnaire (CHAQ), physicians' global assessment of disease activity, parents' global assessment of overall well-being, number of joints with limited ROM, number of active joints, erythrocyte sedimentation rate (30% improvement on 3 measures with <30% deterioration on remaining 3 measures) |
| Weindling et al (2007)[30] | 334 093 | Children with cerebral palsy (n=153/76) | Regular physiotherapy (usual care) plus additional weekly session from physiotherapy assistant for 6 months | Usual care (regular physiotherapy) | Gross Motor Function Measure (14 points) |
| Jolly et al (2007)[22] | 480 612 | People with myocardial infarction or revascularisation (n=450/487) | Home-based self-help manual plus up to 3 face to face and 1 phone call support over 12 weeks | Centre-based cardiac rehabilitation | Incremental shuttle walk test (6 shuttles); Hospital Anxiety and Depression Scale (1.5 points); smoking cessation (20%); blood pressure (6 mmHg systolic); serum cholesterol (0.4 mmol/L) |
| Waterhouse et al (2010)[29] | 460 543 | People with chronic obstructive pulmonary disease (n=372/162) | Twice weekly community-based pulmonary rehabilitation | Twice weekly hospital-based pulmonary rehabilitation | Endurance shuttle walk test (60% increase in distance walked) |
| Glazener et al (2011)[20] | 1 051 699 | Men with incontinence post-prostate surgery (696/788) | Assessment and treatment and exercise over 4 face to face sessions plus advice leaflet | Advice leaflet | Self-reported urinary incontinence (15% reduction in % of people with urinary incontinence) |
| Bowen et al (2012)[17] | 1 457 533 | Adults with aphasia or dysarthria after stroke (n=170/153) | Speech and language therapy visits up to 3 sessions per week for up to 16 weeks | Volunteer visits up to 3 sessions per week for up to 16 weeks | Therapy outcome measure (0.5) |
| Lamb et al (2012)[23] | 755 310 | People with whiplash with persistent symptoms (n=422/507) | 6 sessions of assessment and treatment/exercise over 8 weeks | Single session of advice | Neck Disability Index (3 points) |

Continued

**Table 1** Continued

| Author (year published) | Funding awarded (£) | Population (target sample size/number of participants with primary outcome data) | Intervention | Control | Primary outcome (MCID or % change study aimed to detect) |
|---|---|---|---|---|---|
| Underwood et al (2013)[27] | 1 957 884 | Care home residents (n=409/493) | Twice weekly exercise group for a year | Depression awareness training for care home staff | Geriatric Depression Scale (17.3% reduction in % of people with depression) |
| Logan et al (2014)[24] | 993 080 | People with stroke (n=440/503) | Up to 12 therapy visits to increase outdoor mobility plus verbal/written advice | Verbal/written advice | SF-36 Social function domain (12.5 points) |
| Williams et al (2015)[31] | 976 955 | People with rheumatoid arthritis (n=352/438) | 6 sessions of exercise plus home exercises over 12 weeks | Single assessment advice session with 2 further optional sessions over 12 weeks (no exercises) | Michigan Hand Outcome Questionnaire (0.3) |
| AVERT Group (2015)[21] | 282 372 | People with stroke (n=2104/2083) | 3 additional out of bed sessions per day for up to 2 weeks | Usual care | Modified Rankin Scale (mRS) (7.1% absolute risk reduction of an mRS score of 3–6) |
| Sackley et al (2016)[26] | 1 797 676 | Care home residents with stroke (n=660/870) | Individualised occupational therapy | No occupational therapy | Barthel Index (2 points) |
| Clarke et al (2016)[18] | 1 436 006 | People with Parkinson's (n=680/699) | Up to 8 individualised sessions of Physiotherapy and up to 8 individualised sessions of occupational therapy | No therapy | Nottingham Extended Activities of Daily Living (2.5 points) |

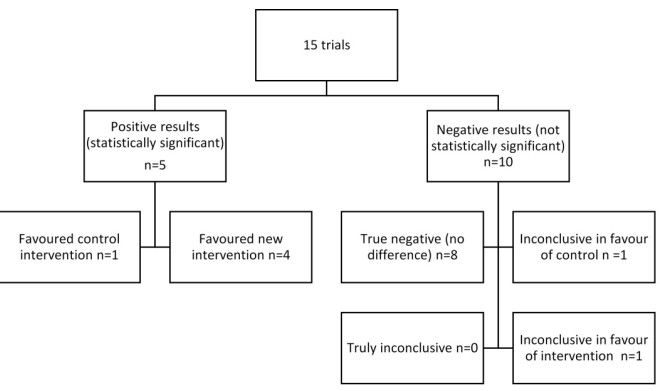

**Figure 2** Classification of primary outcome.

delivered by both a physiotherapist and an occupational therapist. The chief investigators leading the studies were physicians (n=7), physiotherapists (n=5), occupational therapists (n=1), psychologists (n=1) and methodologists (n=1). The total amount of research funding awarded was £12 515 823.

One-third of studies (5/15) reported a definitive finding in favour of one of the treatment arms—four studies in favour of the new treatment, one in favour of the control. Of those with negative results, eight studies reported a true negative (no difference) outcome, one was inconclusive in favour of the new treatment and one inconclusive in favour of the control treatment (figures 2 and 3).

Qualitative data informed 2 themes and 10 subthemes which enabled us to develop data-driven quality ratings:
1. *Preparatory work* (need for the study, underpinning theory for the intervention, co-design, context considerations and intervention piloting).

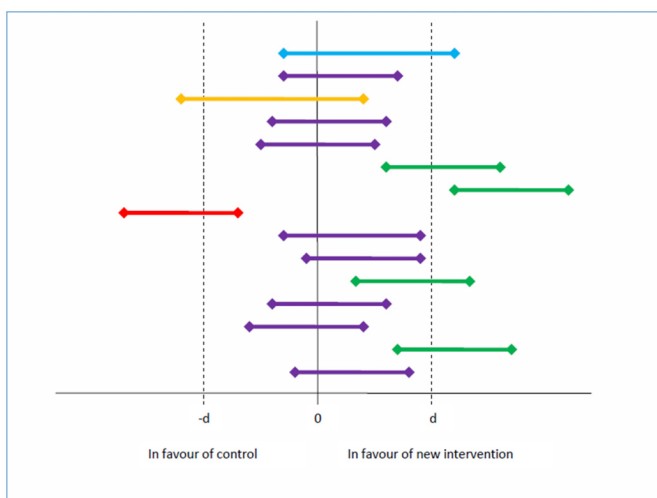

Key: Green = statistically significantly in favour of intervention; Red = statistically significantly in favour of control; Blue = Inconclusive in favour of intervention; Yellow = Inconclusive in favour of control; Purple = True negative (no difference)

**Figure 3** Treatment success of included trials based on 95% CI and minimum clinically important difference from sample size calculation (d).

2. *Intervention and control* (intervention content and dose, individual tailoring, adherence strategies, standardised training, control content and dose).

Table 2 provides examples of summary data underpinning each rating, with table 3 describing the quality rating for each study in chronological order. Table 4 presents the integrative qualitative and quantitative analysis using a joint display. No single study was deemed to be high quality in each subtheme. This said, the two best rated studies reported only expert clinical input into co-designing the intervention with a lack of clear patient and public involvement; however, they reported a definitive trial outcome in favour of the new intervention. There does not appear to be a single aspect of intervention development driving study outcomes. This said, those with lower quality development work appear more likely to show no difference in outcomes compared with those with higher quality development work. Some areas of intervention development appear to be improving with time, these being articulating a clear need and theoretical underpinning, co-design, piloting and descriptions of intervention and control components.

## DISCUSSION

Physical rehabilitation research targets a broad population, although we found that studies for people with stroke to be the most common (n=4). We established that only one-third (5/15) of the RCTs of physical rehabilitation funded by the NIHR HTA programme successfully demonstrated a statistically significant effect for one of the randomised groups in each trial. Four (27%) trials found an effect in favour of the 'new' intervention. Although we would not expect all studies to demonstrate effectiveness in favour of the 'new' intervention, the equipoise principle implies that there would be no difference between the proportion of studies favouring intervention or control.[9] However, this does not account for a null outcome. We were able to use contemporary research methods to develop an assessment of the quality of development work and assessed the included trials to be of varied quality in terms of intervention development work. In general, we found that comprehensive intervention development may have a positive relationship with treatment success. Two studies[23 31] with high quality intervention development reported treatment success, although two older[25 28] and possibly less well reported trials also reported effective interventions. Developments in complex intervention evaluation,[5] reporting standards[16 32] and involving patients and the public in research[33] have occurred since the inception of the HTA programme, and as such, some development work may have been undertaken but not reported in the older studies. A recent overview of approaches to developing interventions noted the absence of patient and public involvement.[34] In addition, there was limited evidence of piloting the intervention prior to proceeding to the full trial with only four studies reporting this having

**Table 2** Description of themes, subthemes and quality ratings with examples

| Theme | Subtheme | Description of rating | Examples of data supporting rating | Rating |
|---|---|---|---|---|
| Preparatory work | **Need for the study** | Multiple sources of evidence of need for the study, for example, recent systematic review, guidelines, high level reports, commissioned research, national audit | International task force highlighted lack of evidence and need for evaluation. Cochrane review drew similar conclusions. | ◀ (green) |
| | | Single source of evidence/non-systematic review to support need for study | Old systematic review indicates paucity of high quality research. | ■ (amber) |
| | | Lack of clarity or underpinning evidence regarding need for study | Poor justification for the study. Evidence cited does not support the need for this particular study. | ● (red) |
| | **Theoretical underpinning** | Theoretical underpinning described | Physiological and psychological theories underpinning the intervention described in detail. | ◀ (green) |
| | | Lacks clear theoretical underpinning | No information provided regarding the theoretical basis for the intervention provided. | ■ (amber) |
| | **Co-design** | Good PPI and expert clinical input | Patients and clinicians helped develop the intervention. | ◀ (green) |
| | | Good PPI but weak or no expert clinical input/Good clinical input but unclear or no PPI | Clinicians contributed to the intervention development but no indication of service user involvement. | ■ (amber) |
| | | No co-design | No co-design was undertaken to develop the intervention. | ● (red) |
| | **Contextual considerations** | Context considered | The use of different professionals in delivering the intervention reflected the real-world situation of how this would occur in practice. | ◀ (green) |
| | | Context not adequately considered | There was a lack of understanding of relevant context and factors needed for intervention development and delivery. | ● (red) |
| | **Piloting of intervention** | Pilot conducted, evaluated and findings addressed for main evaluation | The pilot data helped refine the intervention for evaluation in the main trial. | ◀ (green) |
| | | Pilot conducted but findings not clearly addressed in intervention for main evaluation | The pilot work led to a modification of the control intervention but unclear as to whether this also happened for the novel intervention. | ■ (amber) |
| | | No pilot reported | No piloting of intervention reported | ● (red) |

Continued

**Table 2** Continued

| Theme | Subtheme | Description of rating | Examples of data supporting rating | Rating |
|---|---|---|---|---|
| Intervention and control | **Content and dose** | Intervention components and dose clearly described | The content and the dose of the exercise programme were described in detail. | ◀ High |
| | | Intervention components clearly described but dose was not standardised | The content of the programme was well described but no specific dose was prescribed. | ■ Some/Unclear |
| | | Intervention not replicable from description of components and dose | Intervention was based on usual practice and had no protocol or guidance on minimum dose. | ● Limited |
| | **Tailoring** | Formalised assessment to inform tailoring | An assessment tool was used to determine the individuals level of exercise intensity | ◀ High |
| | | Clinical judgement only used to inform tailoring | Therapists used their clinical judgement to individually tailor programmes. | ■ Some/Unclear |
| | | Not adequately reported | Intervention individually tailored but no information as to how this was undertaken. | ● Limited |
| | **Adherence support strategies** | Explicit strategies to support adherence to the intervention clearly reported | Specific adherence strategies described as part of the intervention. | ◀ High |
| | | No clear information regarding adherence support strategies | No information reported regarding adherence strategies. | ● Limited |
| | | Supporting adherence is not relevant to the intervention | The intervention was passive and adherence strategies not relevant. | NA |
| | **Intervention training** | Standardised training in intervention received +/– additional/ongoing support or training | Staff attended a 1.5-day training session and had an additional support session with ongoing contact from research team. | ◀ High |
| | | No standardised intervention training received but staff delivering described to be experienced in the intervention or training of staff unclear/not reported | Staff have postgraduate training in the intervention but no study-specific training reported. | ■ Some/Unclear |
| | **Control description** | Active control/attention control/usual care with some standardised components | Control was an active intervention that differed from intervention only in terms of delivery setting. | ◀ High |
| | | Usual care had no standardised components | Control was usual care and was not standardised between sites. | ■ Some/Unclear |

Key: ◀ High quality. ■ Some/Unclear quality ● Limited quality.
PPI, Patient and Public Involvement.

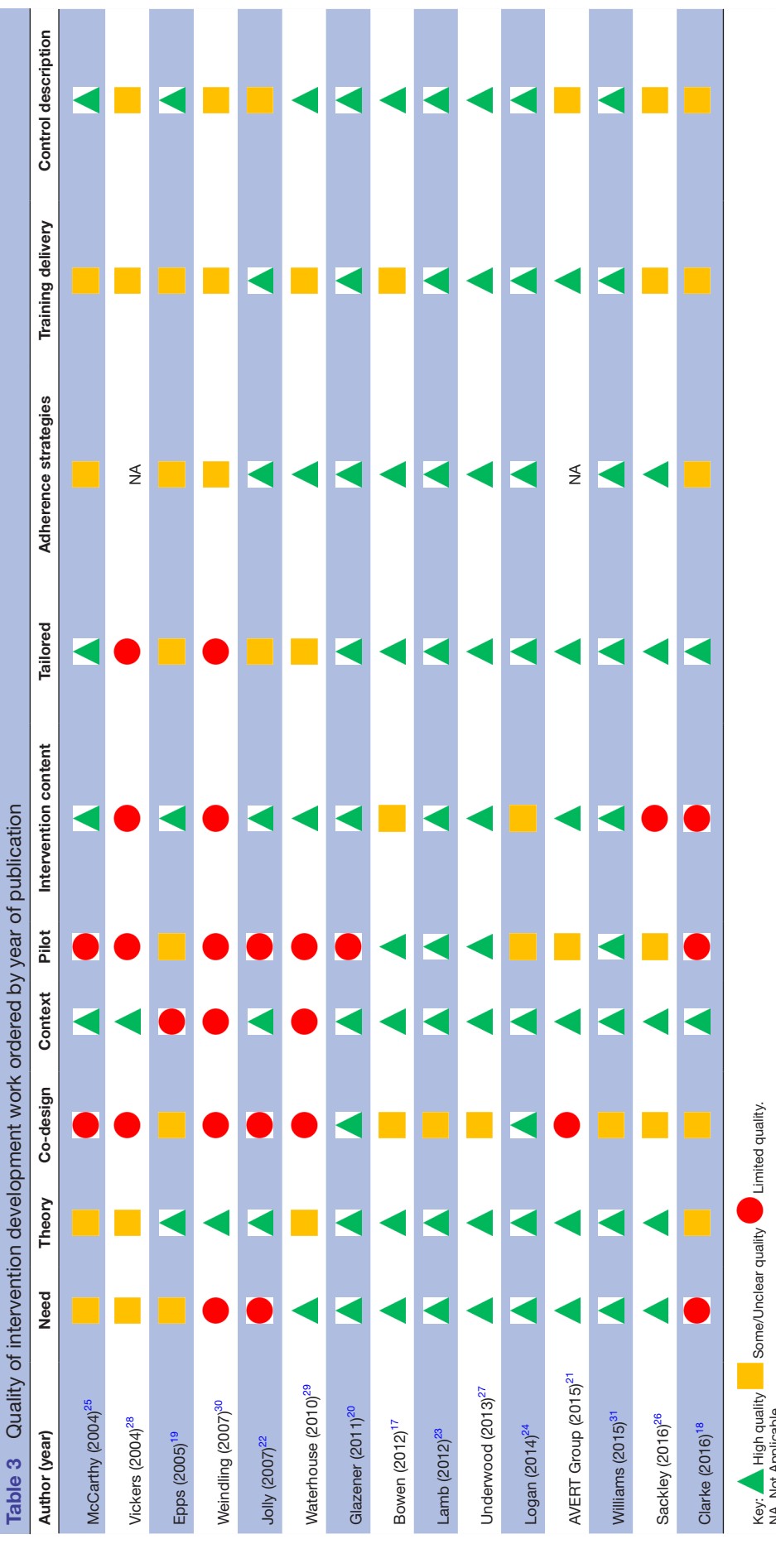

**Table 3** Quality of intervention development work ordered by year of publication

| Author (year) | Need | Theory | Co-design | Context | Pilot | Intervention content | Tailored | Adherence strategies | Training delivery | Control description |
|---|---|---|---|---|---|---|---|---|---|---|
| McCarthy (2004)[25] | 🟧 | 🟧 | 🔴 | ◣ | 🔴 | ◣ | ◣ | 🟧 | 🟧 | ◣ |
| Vickers (2004)[28] | 🟧 | 🟧 | 🔴 | ◣ | 🔴 | 🔴 | 🔴 | NA | 🟧 | 🟧 |
| Epps (2005)[19] | 🟧 | ◣ | 🟧 | 🔴 | 🟧 | ◣ | 🟧 | 🟧 | 🟧 | ◣ |
| Weindling (2007)[30] | 🔴 | ◣ | 🔴 | 🔴 | 🔴 | 🔴 | 🔴 | 🟧 | 🟧 | 🟧 |
| Jolly (2007)[22] | 🔴 | ◣ | 🔴 | ◣ | 🔴 | ◣ | 🟧 | ◣ | ◣ | 🟧 |
| Waterhouse (2010)[29] | ◣ | 🟧 | 🔴 | 🔴 | 🔴 | ◣ | 🟧 | ◣ | ◣ | ◣ |
| Glazener (2011)[20] | ◣ | ◣ | ◣ | ◣ | 🔴 | ◣ | ◣ | ◣ | ◣ | ◣ |
| Bowen (2012)[17] | ◣ | ◣ | 🟧 | ◣ | ◣ | 🟧 | ◣ | ◣ | ◣ | ◣ |
| Lamb (2012)[23] | ◣ | ◣ | 🟧 | ◣ | ◣ | ◣ | ◣ | ◣ | ◣ | ◣ |
| Underwood (2013)[27] | ◣ | ◣ | 🟧 | ◣ | ◣ | ◣ | ◣ | ◣ | ◣ | ◣ |
| Logan (2014)[24] | ◣ | ◣ | ◣ | ◣ | 🟧 | 🟧 | ◣ | NA | 🟧 | 🟧 |
| AVERT Group (2015)[21] | ◣ | ◣ | 🔴 | ◣ | 🟧 | ◣ | ◣ | ◣ | ◣ | ◣ |
| Williams (2015)[31] | ◣ | ◣ | 🟧 | ◣ | ◣ | ◣ | ◣ | ◣ | ◣ | ◣ |
| Sackley (2016)[26] | ◣ | 🟧 | 🟧 | ◣ | 🟧 | 🔴 | ◣ | 🟧 | 🟧 | 🟧 |
| Clarke (2016)[18] | 🔴 | ◣ | 🟧 | ◣ | 🔴 | 🔴 | ◣ | 🟧 | 🟧 | 🟧 |

Key: ◣ High quality 🟧 Some/Unclear quality 🔴 Limited quality.
NA, Not Applicable.

**Table 4** Joint display of treatment success ordered by quality of intervention development work

| Author | Need | Theory | Co-design | Context | Pilot | Intervention content | Tailored | Adherence strategies | Training delivery | Control description | Treatment success |
|---|---|---|---|---|---|---|---|---|---|---|---|
| Lamb[23] | High | High | Some/Unclear | High | High | High | High | High | High | High | Statistically significant in favour of intervention |
| Williams[31] | High | High | Some/Unclear | High | High | High | High | High | High | High | Statistically significant in favour of intervention |
| Underwood[41] | High | High | Some/Unclear | High | High | High | High | High | High | High | True negative (No difference) |
| Glazener[20] | High | High | High | High | Limited | High | High | High | High | High | True negative (No difference) |
| Logan[24] | High | High | High | High | Some/Unclear | Some/Unclear | High | High | High | High | True negative (No difference) |
| Bowen[17] | High | High | Some/Unclear | High | High | Some/Unclear | High | High | Some/Unclear | High | Inconclusive in favour of intervention |
| AVERT Group[21] | High | High | Limited | High | Some/Unclear | High | High | NA | High | Some/Unclear | Statistically significant in favour of control |
| Sackley[26] | High | High | Some/Unclear | High | Some/Unclear | Limited | High | High | Some/Unclear | Some/Unclear | True negative (No difference) |
| Jolly[22] | Limited | High | Limited | High | Limited | High | Some/Unclear | High | High | Some/Unclear | True negative (No difference) |
| McCarthy[25] | Some/Unclear | Some/Unclear | Limited | High | Limited | High | High | Some/Unclear | Some/Unclear | High | Statistically significant in favour of intervention |
| Waterhouse[29] | High | Some/Unclear | Limited | Limited | Limited | High | Some/Unclear | High | Some/Unclear | High | True negative (No difference) |
| Epps[19] | Some/Unclear | High | Some/Unclear | Limited | Some/Unclear | High | Some/Unclear | Some/Unclear | Some/Unclear | High | Inconclusive in favour of control |
| Clarke[18] | Limited | Some/Unclear | Some/Unclear | High | Limited | Limited | Limited | Some/Unclear | Some/Unclear | Some/Unclear | True negative (No difference) |
| Vickers[28] | Some/Unclear | Some/Unclear | Limited | High | Limited | Limited | Limited | NA | Some/Unclear | Some/Unclear | Statistically significant in favour of intervention |
| Weindling[30] | Limited | High | Limited | Limited | Limited | Limited | Limited | Some/Unclear | Some/Unclear | Some/Unclear | True negative (No difference) |

Key: ◀ High quality ◼ Some/Unclear quality ● Limited quality
NA, not applicable.

been done. Most (>80%) drug intervention development studies fail to reach the 'phase III' trial stage.[35] Public health interventions have tended to go straight to an RCT without piloting, which may contribute to challenges in demonstrating effectiveness.[11] There are, of course, other factors that influence trial findings, including trial methods and conduct; however, our question was specifically determined to explore what, if any, relationship existed between intervention development and outcomes and not in the effectiveness of particular interventions.

A strength of our study is the use of integrative mixed methods analysis which has enabled us to explore the relationship between development work and outcome. This rarely used approach in evidence synthesis[36] has given us a unique insight that would not have been possible using a quantitative or qualitative analysis alone. A limitation of our work could be the focus on a single UK funding stream which does not necessarily reflect the body of research funded from other sources, and therefore, the quality of intervention development work is not necessarily generalisable. However, the NIHR HTA programme is the single largest funder of RCTs of applied health research in the UK. They publish detailed monographs of their funded studies, along with protocols and other supporting publications that provide a detailed and rich source of data beyond what would normally be available in journal-based peer-reviewed publications alone. We were able to retain the essence and nuances of the qualitative data while developing categorical ratings of quality to help us better explore the relationship between development work and treatment success.

Our findings are similar to those of Dent and Raftery[9] in relation to those trials showing a benefit who reported 19% (16/85) of studies found in favour of the new intervention. It has been suggested that a 50% success rate is a good investment for healthcare research[37]; however, our findings indicate that the studies we reviewed fell well below this. In contrast, we observed a considerably larger proportion of true negative studies (8/15; 53%) compared with 19/85 (22%) reported by Dent and Raftery.[9] The difference is even greater when compared with a review of cancer trials in the USA where only 2% of trials found a true negative outcome.[10] The reasons for the differences are unclear but could include the pragmatic nature of HTA-funded trials and the relative smaller effect sizes often associated with trials of rehabilitation.[38]

It has been recently suggested that RCTs should only be undertaken if they are justified both scientifically and ethically by having a clear hypothesis and established uncertainty[39] and our findings support that by way of good quality intervention development work. Our findings also align with the elements suggested to be key for developing interventions and reducing research waste by increasing the likelihood of success[40] which will form a comprehensive supplement to the development phase of the updated MRC guidance on developing and evaluating interventions due for publication in 2019. The NIHR HTA is publicly funded and by increasing effort and focus on developing rehabilitation and other interventions in the future, researchers and funding bodies could increase the possibility of a definitive trial reporting beneficial findings after much investment of time and public money.

## CONCLUSIONS

Despite much research effort and funding, only 4 out of 15 evaluations of 'new' rehabilitation interventions funded by the NIHR HTA programme were found to be unequivocally effective. Most studies reported no difference in outcome between study arms. We have used mixed methods research to explore the relationship between intervention development work and treatment success and developed a method of assessing the quality of this work, which suggests comprehensive intervention development work may have a positive relationship with treatment success.

### Recommendations

As this was an exploratory study, further work should be undertaken to establish the validity of quality assessment of intervention development work. This said, researchers and funding agencies should not undervalue the potential benefit of high quality intervention development work prior to definitive RCTs to reduce the likelihood of a null outcome and improve current rates of treatment success.

**Acknowledgements** This work was supported by the National Institute of Health Research (NIHR) Collaboration for Leadership in Applied Health Research and Care South West Peninsula (PenCLAHRC). The views expressed are those of the authors and not necessarily the NHS, the NIHR or the Department of Health and Social Care.

**Contributors** VAG: conception and design, data collection, analysis and interpretation, drafting and approving the manuscript; JJH: design, data collection, analysis and interpretation, drafting and approving the manuscript; JAF: data collection, analysis, revising and approving the manuscript; KF: data collection, revising and approving the manuscript; CP: data collection, revising and approving the manuscript; DAR: conception, revising and approving the manuscript.

**Funding** This study was not funded.

**Competing interests** None declared.

**Patient consent for publication** Not required.

**Provenance and peer review** Not commissioned; externally peer reviewed.

**Data availability statement** No data are available.

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
