## [Reviewer comments · BMJ Open]

ARTICLE DETAILS

TITLE (PROVISIONAL)	Intervention development and treatment success in UK Health Technology Assessment funded trials of physical rehabilitation: a mixed methods analysis
AUTHORS	Goodwin, Victoria; Hill, Jacqueline; Fullam, James; Finning, Katie; Pentecost, C; Richards, David

VERSION 1 – REVIEW

REVIEWER	Anders Hansen Research Unit for Rehabilitation, University of Southern Denmark, Denmark
REVIEW RETURNED	11-Dec-2018

GENERAL COMMENTS	The paper by Goodwin and colleagues presents a review on NIHR THA funded rehabilitation trials and report results of an integrative mixed-method analysis on intervention development and trial-success. The paper is of interest, however, in my opinion, several points need to be better clarified. I have listed some comments below, which may help improve the manuscript. My main concern is that readers are unable to assess data, and thereby the premise, which the analysis and result section is built on. I strongly recommend that the included papers are reported following the PRISMA checklist/flow-chart. There may be major differences in the included trial, which makes interpretation difficult. Without a presentation of the population, intervention, adherence, compliance, intensity, outcome, and outcome measures used, it is difficult to support the conclusion of the paper. A rehabilitation paper using a symptom-based outcome is more likely to succeed, compared to a study of equal methodological quality using quality of life as an outcome. Also, without referral to the included papers, we have no chance of getting inspired by the good examples. It is not clear to me why authors have excluded trials of psychological or cognitive interventions. Please provide a rationale why only 'physical rehab' (PT, OT, speech and language therapists) is used. The literature search must be more specific. It is unclear if authors are including only inter-, and/or multidisciplinary rehabilitation trials, or also include single-profession studies. This is important since the paper is investigating complex interventions. Page 10, line 11-12 authors state: "In general, we found that those studies with better quality intervention development work were more likely to report treatment success". Based on the presented
---

	data this statement is not valid. 50% of studies that were 'Statistically significant in favour of intervention' had limited intervention development quality. In the "strength and limitations of the study" bullets after the abstract, authors write: "factors other than the intervention development can influence treatment success". This is not discussed as a limitation in the discussion section. Figure 3 'treatment success' provides no additional information as the results have been presented in table 2 and figure 2. Minor comments: Please define NIHR in the abstract and MRC in the introduction before abbreviating. It is no clear to me what reference #11 supports in the background section? Page 6 line 3-4: Please assure that the referred trials count up to 15 by inserting the references. Reference list: Please provide full information on reference #5: In the descriptions of the figure; all figures are named Figure 1.
--	--

REVIEWER	Janet Froulund Jensen Holbaek Hospital, Department of Anaesthesiology, Denmark
REVIEW RETURNED	13-Mar-2019

GENERAL COMMENTS	Manuscript ID Review bmjopen-2018-026289 entitled "Intervention development and treatment success in UK Health Technology Assessment funded trials of rehabilitation: a mixed methods analysis" Overall comments: Thank you for the opportunity to review this manuscript. I found your study very interesting and relevant. The authors should be congratulated for doing this work and contributing toward areas of significance for focusing on rehabilitation trials. The fact that rehabilitation trials report conflicting evidence of their effectiveness calls for attention towards investigating the quality of intervention development work as one contextual factor for treatment success. Overall, it is very interesting mixed methods approach and the topic is significant for developing effective interventions in the future. However, the manuscript has to be improved by considering both major and minor comments listed below. BACKGROUND Page 3, line 5-6: there seem to be missing maintaining in this definition as WHO write the following: "...assist individuals who experience, or are likely to experience, disability to achieve and maintain optimal functioning in interaction with their environments". Page 3, line 15-16: Please clarify who developed the framework? Moreover, what framework are you referring to? Page 3, line 33-42: Please rephrase this sentence. Page 3, line 38: Please clarify why you only included RCTs of physiotherapy, occupational therapy and speech and language therapy and why not other healthcare professionals - what about
--

	psychological, cognitive, or social rehabilitation? (or in your paragraph: "Data sources and inclusion criteria") Page 3, line 39: Please invite the unknown reader what the Djulbegovic's classification is, because is it a well-known classification? – A new one? – Or not validated or consensus-based classification? In general, please clarify the rationale for conducting this study. METHODS Design Page 3, line 52-54 and Page 4, line 4: Please add a reference to the funding database. Data sources and inclusion criteria Page 4, line 12: Please clarify why all these exclusion criteria. Page 4, line 18-19: Please rephrase, because all therapies were in one cluster combined with rehabilitation as it could be misunderstood Page 4, line 13, please add (CI) as you use this abbreviation later in the manuscript In general, to enhance the readers' understanding of data sources it is beneficial to describe what is quantitative and what is qualitative data, including the priority in terms of explanatory power i.e., QUAN-qual or? Moreover, it seems to lack a description on how the rating emerged, and the guidelines were analyzed. Please add some ethical committee approval. RESULTS Page 5, line 49: 7834 participants? Please add trial numbers. Page 5/6, line 54-55/1-2: please be consistent in reporting trials meaning that when you report study designs and afterwards the type of rehabilitation. Page 6, line 8: What does multiple professions include? Page 6, line 13: please elaborate why the total amount of research funding is relevant? It seems more interesting to report if they were fully or only partly financed. Page 9, Table 2: please add references, and arrange trials by publication year. In general, please add references to this paragraph. This is highly relevant to at least know which trial that reported positive/negative results. Furthermore, it would be nice to know the following: 1) if the rating system is validated or had undergone an expert evaluation, and 2) an overall rating - or a total score - across themes emerged in the qualitative data. DISCUSSION Page 10, line 32-37: Please rephrase to ease the text for the reader. Page 10, line 45: typo “,,” In general, there seem to miss a general discussion across trials about similarities and differences, for example in target population, setting, intervention or comparator related to existing literature. RECOMMANDATIONS In general, what about trials that included pilot-testing the intervention? That may also influence implementation, quality, and perhaps treatment success. TABLES AND FIGURES Page 14, Figure 1: please add excluded trials with reasons in the first part of the flow diagram.
--	--

	Page 14/15/15: there must be a typo as there are three figure 1. Page 15/16, figure 2 and 3: please add reference if possible. Perhaps, as in Cochrane reviews.
--	---

VERSION 1 – AUTHOR RESPONSE

Reviewer 1	
My main concern is that readers are unable to assess data, and thereby the premise, which the analysis and result section is built on. I strongly recommend that the included papers are reported following the PRISMA checklist/flow-chart. There may be major differences in the included trial, which makes interpretation difficult. Without a presentation of the population, intervention, adherence, compliance, intensity, outcome, and outcome measures used, it is difficult to support the conclusion of the paper. A rehabilitation paper using a symptom-based outcome is more likely to succeed, compared to a study of equal methodological quality using quality of life as an outcome. Also, without referral to the included papers, we have no chance of getting inspired by the good examples.	We have added the references to the included studies to the results section and a table of the population, intervention, control and outcomes for each study (Table 1). Pages 6 and 8 We thank you for your recommendation regarding the PRISMA checklist, however, as our study is not a systematic review and, as such, a number of the checklist items are not relevant.
It is not clear to me why authors have excluded trials of psychological or cognitive interventions. Please provide a rationale why only 'physical rehab' (PT, OT, speech and language therapists) is used.	We and others had noted that an increasing number of large trials of physical rehabilitation were reporting null results, hence the focus on this area. This has been added to the manuscript. Page 3
The literature search must be more specific. It is unclear if authors are including only inter-, and/or multidisciplinary rehabilitation trials, or also include single-profession studies. This is important since the paper is investigating complex interventions.	The interventions could be single or multi-professional and we have clarified this. Page 4
Page 10, line 11-12 authors state: "In general, we found that those studies with better quality intervention development work were more likely to report treatment success". Based on the presented data this statement is not valid. 50% of studies that were 'Statistically significant in favour of intervention' had limited intervention development quality.	We have revised this discussion (which now matches our abstract)
In the "strength and limitations of the study" bullets after the abstract, authors write: "factors other than the intervention development can influence treatment success". This is not	The end of the first paragraph in the discussion did highlight that study conduct could influence treatment success but that this was outside the scope of our research question about intervention

discussed as a limitation in the discussion section.	development. We have revised the text. Pages 2 and 12
Figure 3 'treatment success' provides no additional information as the results have been presented in table 2 and figure 2.	We appreciate your comment but feel this visual representation of the data aids the reader to quickly interpret treatment success across the different studies
Please define NIHR in the abstract and MRC in the introduction before abbreviating.	This has been corrected. Pages 2 and 3
It is no clear to me what reference #11 supports in the background section?	This reference was added in error and has been removed.
Page 6 line 3-4: Please assure that the referred trials count up to 15 by inserting the references.	The references have been added. Page 6
Reference list: Please provide full information on reference #5:	This has been corrected. Page 14
In the descriptions of the figure; all figures are named Figure 1.	This has been corrected.

Reviewer 2	
Page 3, line 5-6: there seem to be missing maintaining in this definition as WHO write the following: ...assist individuals who experience, or are likely to experience, disability to achieve and maintain optimal functioning in interaction with their environments”.	Whilst we appreciate the definition you suggest, the quote we provided was taken directly from the WHO document cited.
Page 3, line 15-16: Please clarify who developed the framework? Moreover, what framework are you referring to?	This has been clarified. Page 3
Page 3, line 33-42: Please rephrase this sentence.	Much of this sentence has been removed or revised to improve readability. Page 3
Page 3, line 38: Please clarify why you only included RCTs of physiotherapy, occupational therapy and speech and language therapy and why not other healthcare professionals - what about psychological, cognitive, or social rehabilitation? (or in your paragraph: “Data sources and inclusion criteria”)	We and others had noted that an increasing number of large trials of physical rehabilitation were reporting null results, hence the focus on this area. This has been added to the manuscript. We have revised the sentence to be clear that this study examined RCTs of physical rehabilitation, rather than focus on the professions that may deliver the interventions. Page 3 and 4
Page 3, line 39: Please invite the unknown reader what the Djulbegovic’s classification is, because is it a well-known classification? – A new one? – Or not validated or consensus-based classification?	To avoid any confusion, we have removed the reference to Djulbegovic here as the categorisation is already described in the analysis section.
In general, please clarify the rationale for conducting this study.	We and others had noted that an increasing number of large trials of physical rehabilitation were reporting null results. There is also a desire to improve intervention development but it isn't clear as to whether this would improve treatment success. This has been added to the manuscript. Page 3

Page 3, line 52-54 and Page 4, line 4: Please add a reference to the funding database.	Under the search and screening section we had previously indicated that the database we searched no longer existed and had been superseded by the NIHR Journal Library for which we have added a web link. Page 4
Page 4, line 12: Please clarify why all these exclusion criteria.	We have added clarifications. Page 4
Page 4, line 13, please add (CI) as you use this abbreviation later in the manuscript	This has been added. Page 4
Page 4, line 18-19: Please rephrase, because all therapies were in one cluster combined with rehabilitation as it could be misunderstood	This has been rephrased to include OR between each keyword. Page 4
In general, to enhance the readers' understanding of data sources it is beneficial to describe what is quantitative and what is qualitative data, including the priority in terms of explanatory power i.e., QUAN-qual or? Moreover, it seems to lack a description on how the rating emerged, and the guidelines were analyzed.	We have now indicated which data were quantitative and which were qualitative in the Data extraction section. Page 4 We did not employ any priority in terms of explanatory power. A description of the process we used to develop the ratings was reported on page 5.
Please add some ethical committee approval.	This study was secondary analysis of publically available study reports and therefore ethical approval was not required.
Page 5, line 49: 7834 participants? Please add trial numbers.	We have added participant numbers to Table 1. Page 8
Page 5/6, line 54-55/1-2: please be consistent in reporting trials meaning that when you report study designs and afterwards the type of rehabilitation.	We have restructured this paragraph. Page 6
Page 6, line 8: What does multiple professions include?	This has been rephrased. Page 6
Page 6, line 13: please elaborate why the total amount of research funding is relevant? It seems more interesting to report if they were fully or only partly financed.	Partial funding is not applicable to this funder. We included the funding amount to illustrate the issue around research waste and that significant amounts of public money are being invested in research and this is mentioned in the background and discussion sections.
Page 9, Table 2: please add references, and arrange trials by publication year.	We discussed at length and trialled various order options (including publication date) for presenting these data and we believe ordering by study quality fits best with the aims of our study. Table 1 has the studies in date order. Page 8
In general, please add references to this paragraph. This is highly relevant to at least know which trial that reported positive/negative results. Furthermore, it would be nice to know the following: 1) if the rating system is validated or had undergone an expert evaluation, and 2) an overall rating - or a total score - across themes emerged in the qualitative data.	We have added references to the text and Table 3. Pages 8 and 12 The rating system was developed from the data from our exploratory study and has not undergone further evaluation. However, we would like to assess validity with other trials of complex interventions in the future.

	We discussed at length whether or not to have an overall quality score but we felt that a single score derived from each criteria would imply that each item should be weighted the same and that this may be inappropriate. We opted for a visual representation, based on the Cochrane Risk of Bias tool, to give an indication of quality across a number of items, which would be lost if quality was based on a total score.
Page 10, line 32-37: Please rephrase to ease the text for the reader.	We have revised this very long sentence into two smaller and more concise sentences. Page 12
Page 10, line 45: typo “,,,”	The extra ‘,’ has been removed.
In general, there seem to miss a general discussion across trials about similarities and differences, for example in target population, setting, intervention or comparator related to existing literature.	The purpose of this study was not to examine effectiveness in a homogeneous sample as would be expected in a systematic review and therefore a discussion of PICO characteristics would not add to our synthesis. However, in line with previous comments, we have now included Table 1 to describe these characteristics. Page 8
In general, what about trials that included pilot-testing the intervention? That may also influence implementation, quality, and perhaps treatment success.	We absolutely agree and piloting was one of areas we looked at in terms of quality development work. Unfortunately only 4 of the studies had piloted the intervention (see Table 3). We have included some text regarding piloting in the discussion. Page 12
Page 14, Figure 1: please add excluded trials with reasons in the first part of the flow diagram.	The number of excluded studies has been added to Figure 1 and wording changed to reflect search keywords rather than filters.
Page 14/15/15: there must be a typo as there are three figure 1.	This error has been corrected.
Page 15/16, figure 2 and 3: please add reference if possible. Perhaps, as in Cochrane reviews.	We have now added the references to Table 3 to aid the reader. As Figure 3 is not a Forrest plot we did not feel the addition of references in addition to Table 3 would be useful

VERSION 2 – REVIEW

REVIEWER	Anders Hansen Spine Centre of Southern Denmark, Lillebaelt Hospital, Middelfart, Denmark.
REVIEW RETURNED	06-May-2019

GENERAL COMMENTS	The authors have improved their manuscript according to the reviewers' suggestions. However, after changes were incorporated, I have some old and new questions and concerns, some more serious than other.  At page 4 you state that trials with a lack of clear primary outcome and/or time point are excluded, however, at P5 you define how to deal with not explicitly identified outcomes or time point. Which is correct?
--

	 • P5 You state that you will extract quantitative data including MCID, planned sample size and the amount of founding awarded for each trial. I am unable to find the information. • P7: You state that some areas of intervention development appear to be improving with time. It is difficult to conclude from the table 3, as the studies are not arranged by year. Further, how do you define improvement? What justifies an improvement in co-design? One of the two studies with high quality is below the median regarding the year of publication. • There seems to be error in the referring of trials when comparing table #1 and #3. Table #1 refers to trials by Lawson et al. 2010 and Longhorn et al. 2015. I cannot find these trials in the reference list. Table #3 refers to trials by AVERT and Waterhouse, but these are not presented in table #1 Table #1 refers to trials by Lawson et.al. 2010 and Longhorn et al.2015, but these are not presented in table #3.  • Page 12 L1, please insert physical. • P12, L16: Although I sympathize with the purpose, based on the presented information I still disagree with the authors' statement that poorer quality intervention development work is less likely to report treatment success. If looking at trials over and under the median in terms of quality, I don't find justification for the statement. Please elaborate on how one trial justifies this definitive statement. Higher Quality/Poorer quality Statistically significant in favor of intervention 2, 2 Statistically significant in favor of control 1, 0 Truly negative 3, 4 Inconclusive in favor of the intervention 1, 0 Inconclusive in favor of the control 0, 1 Rephrasing all of this as "comprehensive intervention development may have a positive relationship with treatment success" as the authors did in their final conclusion section is far more appropriate.
--	--

REVIEWER	Janet Froulund Jensen Dept. of Anaesthesia, Holbaek Hospital, Smaedelundsgade 60, 4300 Holbaek, Region Zealand, Denmark.
REVIEW RETURNED	22-May-2019

GENERAL COMMENTS	Manuscript ID Review bmjopen-2018-026289.R1 entitled "Intervention development and treatment success in UK Health Technology Assessment funded trials of rehabilitation: a mixed methods analysis" Overall comments: Thank you for the opportunity to review the revisited version of the manuscript. The authors have done a great job improving the manuscript. Some few minor comments that need to be amended. TITLE Please consider if the title should reflect trials of physical rehabilitation
---

	METHODS Data extraction Page 4, line 55: Please explain abbreviation before using them first time RESULTS Please report studies in the three tables in same order DISCUSSION Please comment on the population of the included studies, because six of the included studies were Neurological (If stroke is included here). Some other considerations are sample size and used outcome measures – did the trials reach sample size? - And were outcome measurements sufficient in the included studies?
--	--

VERSION 2 – AUTHOR RESPONSE

Reviewer: 1	
At page 4 you state that trials with a lack of clear primary outcome and/or time point are excluded, however, at P5 you define how to deal with not explicitly identified outcomes or time point. Which is correct?	This has been clarified. The text under exclusion criteria has been removed.
P5 You state that you will extract quantitative data including MCID, planned sample size and the amount of funding awarded for each trial. I am unable to find the information.	This information has been added to Table 1
P7: You state that some areas of intervention development appear to be improving with time. It is difficult to conclude from the table 3, as the studies are not arranged by year. Further, how do you define improvement? What justifies an improvement in co-design? One of the two studies with high quality is below the median regarding the year of publication.	We have included an additional Table that is ordered by year of publication (new Table 3). An example of quality rating using co-design has been added to the data analysis section.
There seems to be error in the referring of trials when comparing table #1 and #3. Table #1 refers to trials by Lawson et al. 2010 and Longhorn et al. 2015. I cannot find these trials in the reference list. Table #3 refers to trials by AVERT and Waterhouse, but these are not presented in table #1 Table #1 refers to trials by Lawson et.al. 2010 and Longhorn et al.2015, but these are not presented in table #3.	Many thanks for this. The confusion has arisen from the name of the Chief Investigator vs the lead author on the main publication as these have differed for some studies. We have now referred to the study in relation to lead author and modified Table 1.

Page 12 L1, please insert physical.	This has been added.
P12, L16: Although I sympathize with the purpose, based on the presented information I still disagree with the authors' statement that poorer quality intervention development work is less likely to report treatment success. If looking at trials over and under the median in terms of quality, I don't find justification for the statement. Please elaborate on how one trial justifies this definitive statement. Rephrasing all of this as "comprehensive intervention development may have a positive relationship with treatment success" as the authors did in their final conclusion section is far more appropriate.	Thank you for this. We have reworded as suggested.

Reviewer: 2	
Thank you for the opportunity to review the revisited version of the manuscript. The authors have done a great job improving the manuscript. Some few minor comments that need to be amended.	Thank you for your positive comments
TITLE Please consider if the title should reflect trials of physical rehabilitation	The title has been amended.
METHODS Data extraction Page 4, line 55: Please explain abbreviation before using them first time	CI had been explained under the Data sources and inclusion criteria section earlier on the same page.
RESULTS Please report studies in the three tables in same order	We have retained Table 1 in order of publication and added a new table 3 also ordered by publication in relation to Reviewer 1's comments. Table 2 is in relation to themes and does not relate to individual studies. However, we have kept the joint display table (previously table 3, now table 4) ordered by intervention development quality. We have reworded the Table title to provide greater clarity

DISCUSSION Please comment on the population of the included studies, because six of the included studies were Neurological (If stroke is included here). Some other considerations are sample size and used outcome measures – did the trials reach sample size? - And were outcome measurements sufficient in the included studies?	We have added text to the discussion about the population. We have added text to the results section and table 1 about sample size. However, we had already stated in the discussion (end of paragraph 1) that the aim of this review wasn't about trial methods and conduct and their impact on effectiveness. We would see sample size as being part of trial conduct and outside the scope of the study.
---	---

VERSION 3 – REVIEW

REVIEWER	Anders Hansen University of Southern Denmark, Denmark
REVIEW RETURNED	22-Jul-2019

GENERAL COMMENTS	Thank you for the corrections made. 1) Table 3. Please insert the year of the trials to make it easier for the reader to follow. 2) Result section: "The combined target sample size was 7548 participants, 7834 of whom provided primary outcome data" How is that possible? Further, in the previous version, you wrote that the combined sample size was 9035? Also, in the previous version, you wrote that "The total amount of research funding awarded was £11,361,182." Now you state that: "The total amount of research funding awarded was £12,515,823." Please elaborate, and please mark corrections.
--

VERSION 3 – AUTHOR RESPONSE

Reviewer comment	Author response
1) Table 3. Please insert the year of the trials to make it easier for the reader to follow.	The year has been added to Table 3
2) Result section: "The combined target sample size was 7548 participants, 7834 of whom provided primary outcome data" How is that possible? Further, in the previous version, you wrote that the combined sample size was 9035?	We have clarified the text. The reason for the higher combined sample in the original version was because we had included the proposed recruitment target that included an inflation for potential loss to follow up. We felt that this was confusing when we were revising Table 1 as we had in fact 3 different numbers – (a) proposed sample size, (b) proposed recruitment

	target, (c) number with primary outcome. We concluded that a) and c) were the most relevant to our paper.
3) Also, in the previous version, you wrote that "The total amount of research funding awarded was £11,361,182." Now you state that: "The total amount of research funding awarded was £12,515,823." Please elaborate, and please mark corrections.	Apologies, our original calculation was incorrect and we missed highlighting it as a correction.